# Orthosteric and allosteric modulation of human HCAR2 signaling complex

Chunyou Mao [1,2,3,9] ✉, Mengru Gao[4,5,9], Shao-Kun Zang[1,9], Yanqing Zhu[1,9], Dan-Dan Shen [1,6], Li-Nan Chen[1,7], Liu Yang[4], Zhiwei Wang[4], Huibing Zhang [3], Wei-Wei Wang[3], Qingya Shen [3], Yanhui Lu[8] ✉, Xin Ma [4,5] ✉ & Yan Zhang [1,2,3,6] ✉

Hydroxycarboxylic acids are crucial metabolic intermediates involved in various physiological and pathological processes, some of which are recognized by specific hydroxycarboxylic acid receptors (HCARs). HCAR2 is one such receptor, activated by endogenous β-hydroxybutyrate (3-HB) and butyrate, and is the target for Niacin. Interest in HCAR2 has been driven by its potential as a therapeutic target in cardiovascular and neuroinflammatory diseases. However, the limited understanding of how ligands bind to this receptor has hindered the development of alternative drugs able to avoid the common flushing side-effects associated with Niacin therapy. Here, we present three high-resolution structures of HCAR2-Gi1 complexes bound to four different ligands, one potent synthetic agonist (MK-6892) bound alone, and the two structures bound to the allosteric agonist compound 9n in conjunction with either the endogenous ligand 3-HB or niacin. These structures coupled with our functional and computational analyses further our understanding of ligand recognition, allosteric modulation, and activation of HCAR2 and pave the way for the development of high-efficiency drugs with reduced side-effects.

Hydroxycarboxylic acids are indispensable metabolic intermediates that not only regulate energy and lipid metabolism, but also act as signaling molecules to regulate immune, neurological, and other physiological processes[1,2]. Furthermore, these molecules have been implicated in a range of diseases, such as type 2 diabetes, obesity, inflammation, cancer, and neurodegenerative disorders[3,4]. Lactate, β-hydroxybutyrate (3-HB), and 3-hydroxyoctanoic acid are examples of hydroxycarboxylic acids predominantly recognized by a class of G protein-coupled receptors (GPCRs) called hydroxycarboxylic acid

receptors (HCARs), which include HCAR1 (also known as GPR81), HCAR2 (GPR109A), and HCAR3 (GPR109B)[5].

Among these receptors, HCAR2 is of particular interest due to its recognition of the important endogenous metabolites 3-HB and butyrate, plays a role as an anti-atherogenic sclerosing agent and maintains intestinal health, as well as its targeting by the anti-antilipolytic drug Niacin[6–8]. The receptor is highly expressed in adipocytes, immune cells, and epithelial cells, where it exerts its effects on lipolysis and cellular immunity via the Gi/o protein signaling

[1]Department of Biophysics and Department of Pathology of Sir Run Run Shaw Hospital, Zhejiang University School of Medicine, Hangzhou 310058, China. [2]Center for Structural Pharmacology and Therapeutics Development, Sir Run Run Shaw Hospital, Zhejiang University School of Medicine, Hangzhou 310016, China. [3]Liangzhu Laboratory, Zhejiang University Medical Center, Hangzhou 311121, China. [4]School of Medicine, Jiangnan University, Wuxi 214122, China. [5]School of Pharmaceutical Sciences, Jiangnan University, Wuxi 214122, China. [6]MOE Frontier Science Center for Brain Research and Brain-Machine Integration, Zhejiang University School of Medicine, Hangzhou 310058, China. [7]Department of General Surgery, Sir Run Run Shaw Hospital, Zhejiang University School of Medicine, Hangzhou 310016, China. [8]School of Nursing, Peking University, 100191 Beijing, China. [9]These authors contributed equally: Chunyou Mao, Mengru Gao, Shao-Kun Zang, Yanqing Zhu. ✉e-mail: maochunyou@zju.edu.cn; luyanhui@bjmu.edu.cn; maxin@jiangnan.edu.cn; zhang_yan@zju.edu.cn

pathway[5]. Importantly, HCAR2 stands as a potential therapeutic target for neuroimmune disorders, cardiovascular diseases and cancers[4,9,10]. However, the pursuit of drug discovery targeting HCAR2 is frequently hampered by common adverse effects, such as flushing[11]. To overcome this limitation, potent agonists of HCAR2 (such as MK-6892 and MK-1903) with reduced flushing effects have been developed in recent years, and certain allosteric modulators (such as compound 9n) targeting the receptor have also been discovered[12–17]. Nevertheless, their mode of recognition mode and action mechanisms of action remain poorly understood, which hinders further development for efficient pharmacological modulation.

In this study, we applied single-particle cryo-electron microscopy (cryo-EM) to determine the structures of human HCAR2-Gi complexes bound to distinct ligands. These structures, together with pharmacological profiling and computational analyses, provide a thorough and in-depth understanding of the ligand recognition and activation mechanisms of HCAR2, broadening our knowledge of the ligand binding mode of GPCRs and facilitating the development of drugs targeting HCA receptors.

## Results

### Overall structure of the HCAR2–Gi1 signaling complex

In accordance with prior studies[14,18,19], our NanoBiT-Gi1 dissociation assay demonstrated that MK-6892 is the most potent agonist of HCAR2, exhibiting a potency of 80 nM (Supplementary Fig. 1a), while Niacin showed strong small-molecule carboxylic acid agonist activity with a potency of 117 nM (Supplementary Fig. 1c). However, the endogenous 3-HB exhibited weak activity with a potency of 0.9 mM (Supplementary Fig. 1e). To facilitate the cryo-EM study of stable HCAR2-Gi1 complexes bound to these distinct ligands, we used several

methods to assemble the complexes, including N-terminal fusion with BRIL protein (the thermostabilized apocytochrome b562a) to increase receptor expression, dominant-negative Gαi1 (DNGαi1), the Gi/o stabilizing antibody scFv16[20], and the NanoBiT tethering strategy[21–23]. Subsequently, the HCAR2-Gi1 complexes were co-expressed in sf9 cells and purified to homogeneity for cryo-EM analysis (Supplementary Figs. 1–4). The structures of HCAR2-Gi1 complexes bound to 3-HB and compound 9n, Niacin and compound 9n, and MK-6892 alone were determined to nominal global resolutions of 2.60 Å, 2.55 Å, and 2.76 Å, respectively, by single-particle cryo-EM, respectively (Supplementary Fig. 2–4, Supplementary Fig. 5a, b and Supplementary Table 1). The high-resolution structures allow us to confidently model the D8 to N302 residues of the receptor and most residues of the Gil trimer (Fig. 1a). Notably, clear densities were observed for all ligands, including 3-HB, Niacin, MK-6892, and compound 9n (Fig. 1b). Interestingly, extra densities were also observed at the N-terminal N17 glycosylation site and modeled with N-linked glycan (Fig. 1c)[24].

The three resolved structures exhibited a similar overall conformation, displaying a comparable Gil coupling interface between three intracellular loops (ICLs) and the intracellular ends of transmembrane helices (TMs) 2/3/5/6/7 of the receptor with the α5 helix, αN and β2–β3 loop of the Gαi1 subunit (Supplementary Fig. 5c–e and Supplementary Fig. 6a, b). The coupling mode of the Gi1 is relatively similar to that of other class A GPCRs[25–27]. However, unlike canonical class A receptors, HCAR2 exhibits a compact global conformation, with the extracellular side completely enclosed by ECL2 (extracellular loop 2) and the N-terminus of the receptor (Fig. 1c). Furthermore, in addition to the conserved disulfide bond pair between ECL2 and the extracellular end of TM3 (Cys100$^{3.25}$–Cys177$^{ECL2}$; superscripts refer to Ballesteros–Weinstein numbering[28]), the N-terminus of HCAR2 forms

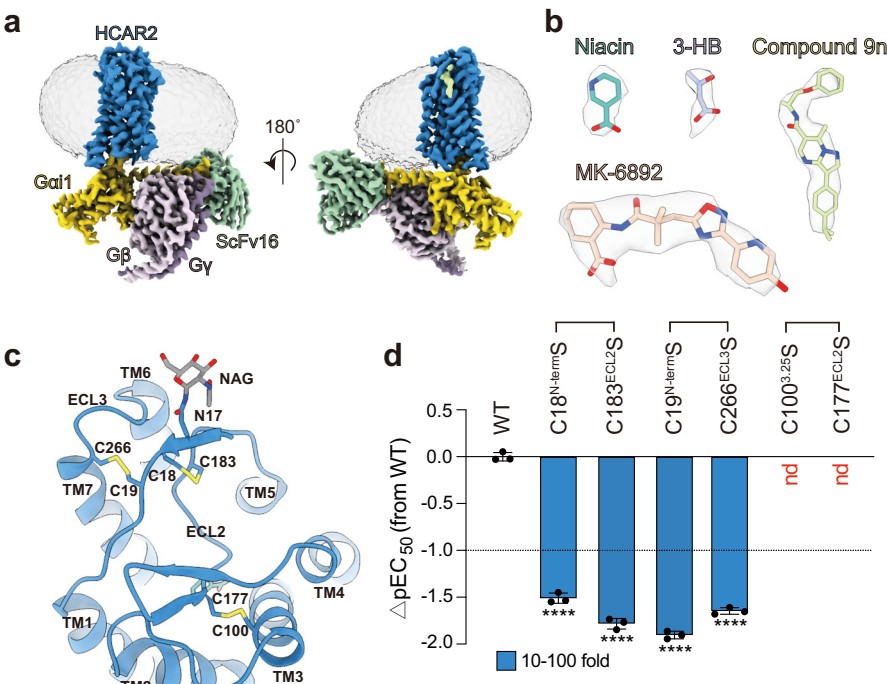

**Fig. 1 | Cryo-EM structure of HCAR2–Gi1 signaling complex. a** Cryo-EM density map of HCAR2–Gi1 complex (represented by the 3-HB–bound complex). HCAR2 is depicted in blue, Gαi1 in yellow, Gβ in light pink, Gγ in dark purple, scFv16 in light green. **b** Cryo-EM densities and models of distinct agonists (Niacin, 3-HB, MK-6892 and compound 9n) from their activated complexes. Niacin in cyan, 3-HB in lavender, MK-6892 in beige, compound 9n in light green. **c** The compact conformation of the extracellular side of HCAR2 is stabilized by three unique pair disulfide bonds (Cys18$^{N-term}$–Cys183$^{ECL2}$, Cys19$^{N-term}$–Cys266$^{ECL3}$, Cys100$^{3.25}$–Cys177$^{ECL2}$). **d** Effects of mutations in the three disulfide bonds on Niacin-induced Gi1 dissociation signal as

indicated by NanoBiT assay. Bars represent differences in calculated agonist potency (pEC50) for each mutant relative to the wild-type receptor (WT). Data are colored according to the extent of effect. nd not determined; ****$P < 0.0001$ (one-way ANOVA followed by Dunnett's multiple comparison test, compared with the response of WT, the detailed $P$ value for each condition is $P < 0.0001$, $P < 0.0001$, $P < 0.0001$, and $P < 0.0001$, from left to right. Data are shown as the mean ± SEM from $n = 3$ independent experiments). Source data are provided as a Source Data file.

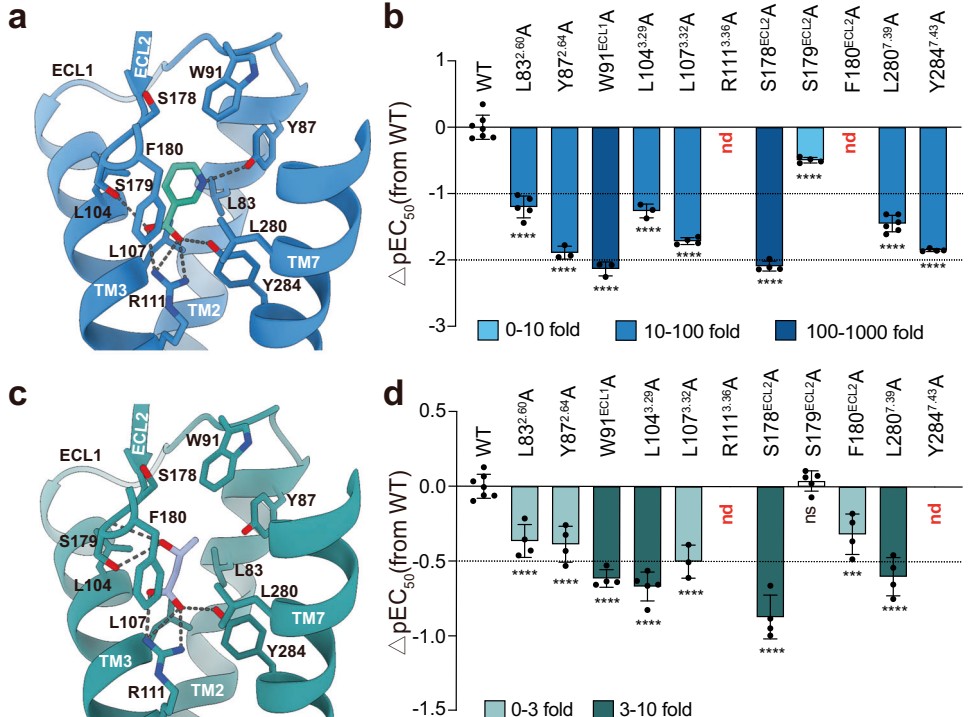

**Fig. 2 | Recognition of small endogenous 3-HB and Niacin drug.** Detailed interactions of Niacin (**a**) and 3-HB (**c**) with HCAR2. Hydrogen bonds are depicted as black dashed lines. The structures of Niacin- and 3-HB-bound HCAR2 include the binding of compound 9n. Effects of mutations in Niacin- (**b**) and 3-HB- (**d**) binding pockets on their induced Gi1 dissociation signals. Bars represent differences in calculated agonist potency (pEC50) for each mutant relative to the wild-type receptor (WT). Data are colored according to the extent of effect. nd, not determined; $^{ns}P > 0.05$; $^{***}P < 0.001$; $^{****}P < 0.0001$ (one-way ANOVA followed by Dunnett's multiple comparison test, compared with the response of WT. Data are shown as the mean ± SEM from three independent experiments). Source data are provided as a Source Data file.

two disulfide bond pairs with ECL2 and ECL3 (Cys18[N-ter]–Cys183[ECL2]; Cys19[N-ter]–Cys266[ECL3]) (Fig. 1c). Sequence analysis suggested that these three disulfide bond pairs are conserved in HCARs (Supplementary Fig. 6c), implying their critical functions[29]. In support of this, mutations in any residue within these three disulfide bonds significantly compromised HCAR2 activation (Fig. 1d and Supplementary Fig. 6d). Thus, these disulfide bonds likely play a role in the ligand binding and activation of HCARs, possibly by stabilizing the extracellular conformation of the receptor.

**Recognition of small endogenous 3-HB and Niacin**
The compact conformation of HCAR2 defines a small elliptical orthosteric binding pocket, which is mainly composed of the TMs 2/3/ 7 and ECLs 1/2 (Fig. 2a, c). High-resolution densities of endogenous Niacin and 3-HB, coupled with detailed structural analysis defined the favorable binding poses for 3-HB and Niacin, which were further validated by docking analysis (Fig. 2a, c, Supplementary Fig. 7a, b, e). Detailed examination of the structures revealed that the carboxyl group of both Niacin and 3-HB inserts into the bottom of the pocket, forming a strong electrostatic interaction with R111[3.36] of the receptor (Fig. 2a, c). Sequence analysis showed that the residue is conserved in HCARs (Fig. 3e), suggesting its importance in carboxyl group recognition[5]. In line with this, our signaling analysis and previous reports revealed that mutations in R111[3.36] abolish both 3-HB- and Niacin-induced downstream signaling (Fig. 2b, d, Supplementary Fig. 7f, Supplementary Data 1), emphasizing the crucial role of R111[3.36] in ligand recognition. In addition to electrostatic interaction, the carboxyl group was further stabilized by additional hydrogen bonds with the Y284[7.43] and S179[ECL2] (Fig. 2a, c). Interestingly, our signaling assays indicated that the mutation of Y284[7.43] but not S179[ECL2] significantly impaired receptor activation (Fig. 2b, d, Supplementary Fig. 7f,

Supplementary Data 1), indicating the more important role of Y284[7.43] in ligand binding and recognition.

The hydroxyl group in 3-HB primarily forms hydrogen bonds with side chain and the main chain of S179[ECL2] (Fig. 1c). Unlike 3-HB, Niacin has no hydroxyl group but preserves a pyridinic nitrogen (Fig. 1a). Our docking analysis indicated that the pyridinic nitrogen of Niacin probably forms a hydrogen bond with Y87[2.64] (Supplementary Fig. 7a). Consistently, replacing Y87[2.64] with F87[2.64] to disrupt the interactions decreased the Niacin potency by more than 10-fold (Supplementary Fig. 7c). In addition to these ionic and polar interactions, extensive hydrophobic and van der Waals interactions, involving L83[2.60], W91[ECL1], L104[3.29], L107[3.32], F180 [ECL2] and L280[7.39], also participate in ligand recognition for both 3-HB and Niacin (Fig. 2c, d, Supplementary Fig. 7f, Supplementary Data 1). However, compared to 3-HB, these residues form a more robust hydrophobic stacking interaction with the pyridine ring of Niacin (Fig. 2a, c, Supplementary Fig. 7f and Supplementary Data 1), greatly enhancing the binding of Niacin to the receptor[19]. Consistent with this, our molecular dynamics (MD) simulations analysis showed that the 3-HB binding exhibited a higher root mean square deviation (RMSD) value (-1.6 Å) and easily escaped from the crevice formed by the TM4-TM5-ECL2 region when compared to that of Niacin (RMSD: -0.7 Å) (Supplementary Figs. 7d and 9b), indicating a more stable binding of Niacin Collectively, our results showed that the binding modes of 3-HB and Niacin to HCAR2 are highly conserved, with similar molecular size and almost identical interacting residues. Relative to 3-HB, the pyridine ring of Niacin plays a role in ligand affinity and potency by enhancing interactions with surrounding residues.

The small size of Niacin easily evokes the endogenous ligands of aminergic receptors, such as serotonin, dopamine and histamine. Structural comparison of these receptors revealed that the 7TM topology of HCAR2 is very similar to that of these aminergic receptors

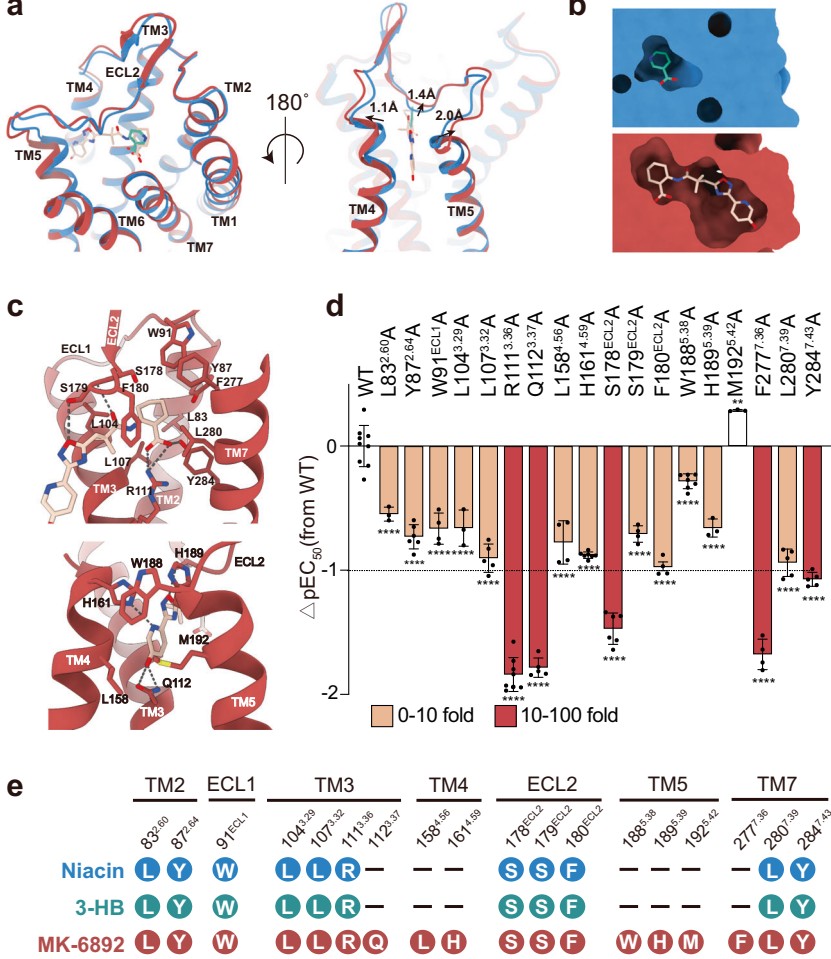

**Fig. 3 | Conformational changes and recognition upon MK-6892 binding.**
**a** Structural comparison of HCAR2 bound to Niacin/compound 9n (blue) and MK-6892(red). **b** Vertical cross-section of the Niacin and MK-6892 binding pocket in HCAR2. **c** Detailed interactions between MK-6892 and HCAR2. Hydrogen bonds are depicted as black dashed lines. **d** Effects of mutations in the MK-6892−binding pocket on its induced Gi1 dissociation signal. Bars represent differences in calculated agonist potency (pEC50) for each mutant relative to the wild-type receptor (WT). **P = 0.0032; ***P < 0.0001 (one-way ANOVA followed by Dunnett's multiple comparison test, compared with the response of WT. Data are shown as the mean ± SEM from n independent experiments, the number of experiments (n) for each condition is 9, 3, 6, 4, 3, 5, 8, 5, 4, 7, 6, 4, 7, 3, 3, 4, 5, 5, and 4, from left to right). Source data are provided as a Source Data file. **e** Comparison of the residues involved in the recognition of the orthosteric agonizts Niacin, 3-HB and MK-6892. Source data are provided as a Source Data file.

(Supplementary Fig. 8a, b)[30–32]. However, unlike aminergic receptors, the extracellular region of HCAR2 is closed by its N-terminus and ECL2, forming a compact configuration (Fig. 1c). As a result, small ligands, including 3-HB and Niacin, are fully enclosed within the binding pocket of HCAR2, rather than being exposed to solvents as in aminergic receptors (Fig. 2a, c, Supplementary Fig. 8a, b)[30–32]. While the endogenous ligands of aminergic receptors primarily interact with TMs 3/5/6/7 and bind deeper within the pocket, the Niacin and 3-HB in HCAR2 are situated closer to TM2 and interacts with TMs 2/3/7 and ECLs 1/2 of the receptor (Supplementary Fig. 8a). Consequently, the key residue R111[3.36], responsible for carboxyl group recognition, occupies a position equivalent to that of the NH2 group of monoamine ligands in aminergic receptors (Supplementary Fig. 8b). In addition, the highly conserved toggle switch W[6.48] within the aminergic receptors is substituted with the relatively smaller F[6.48] in HCAR2 and HCAR3 and Y[6.48] in HCAR1 (Supplementary Fig. 8c). Structural analysis revealed that replacing F244[6.48] with the bulkier W would cause spatial hindrance with R111[3.36] in HCAR2, impeding the recognition of the ligand by R111[3.36] (Supplementary Fig. 8b). Our functional analysis also revealed that mutating F244[6.48] to W[6.48] in HCAR2 significantly decreased the agonist efficacy by 2-fold and the potency by 10-fold, whereas the mutation to Y[6.48] had negligible effects (Supplementary Fig. 8d, e).

Interestingly, the substitution of F244[6.48] with the smaller alanine apparently compromised receptor activation as well (Supplementary Fig. 8d, e). Therefore, a properly sized F/Y[6.48] residue is critical for ligand recognition and the activation of HCARs.

**Recognition and conformational changes upon MK-6892 binding**

Given the pivotal physiological and pathological roles of HCAR2, multiple synthetic agonists have been developed, such as MK-6892 and MK-1903[14,17,33]. Of these compounds, MK-6892 (2-[[3-[3-(5-hydroxypyridin-2-yl)−1,2,4-oxadiazol-5-yl]−2,2-dimethylpropanoyl]amino] cyclohexene-1-carboxylic acid) stands out as the most potent and selective agonist of HCAR2, and exhibits a marked obvious reduction in the flushing side-effects caused by Niacin drugs[14]. Intriguingly, MK-6892 has a much larger chemical scaffold than 3-HB and Niacin, so how can it fit into the limited ligand-binding pocket of HCAR2? Structural comparisons of Niacin- and MK-6892-bound HCAR2 revealed that the binding of MK-6892 induces a notable conformational change in the TM4-ECL2-TM5 region of the receptor, including an upward movement of the central region of ECL2 by 1.4 Å (measured at the Cα atoms of S179[ECL2]) and an outward movement of TM4 and TM5 by 1.1 Å and 2.0 Å respectively (measured at the Cα atoms of H161[4.59] and H189[5.39])

(Fig. 3a). These conformational changes create a significantly larger ligand pocket (639.5 Å³) than that of the endogenous ligand pockets (3-HB: 230.0 Å³) that accommodates the bulkier MK-6892 (Fig. 3b).

Specifically, the cyclohexanecarboxyl group of MK-6892 binds to a position analogous to that of Niacin, while its hydroxypyridine group extends from the cleft between TM4 and TM5 (Fig. 3a, c). The cyclohexanecarboxylic group forms interactions that are similar to those of Niacin (Fig. 3c). Of note, our functional analysis showed that mutations of these residues also decreased MK-6892-induced receptor activation, but to a significantly lesser extent than that of Niacin (Fig. 3d, Supplementary Fig. 9a and Supplementary Data 1). Among these residues, R111$^{3.36}$ was the most significant, with the R111A$^{3.36}$ mutation only reducing the activation of MK-6892 but completely eliminating the activity of Niacin (Figs. 2b, 3d, Supplementary Figs. 7f, 9a and Supplementary Data 1). Furthermore, MK-6892 also forms extensive additional hydrophilic and hydrophobic interactions with HCAR2 (Fig. 3c). Four hydrogen bonds were observed: the carbonyl group with the main chain of S179$^{ECL2}$, the oxadiazole group with the side chain of S179$^{ECL2}$, and the hydroxypyridine with Q112$^{3.37}$ and H161$^{4.59}$ (Fig. 3c). Meanwhile, additional residues on TM4 and TM5 also participate in the unique recognition of MK-6892 when compared to Niacin (Fig. 3c). Our signaling assays demonstrated that mutations in most of these residues significantly impair MK-6892 activation (Fig. 3d, Supplementary Fig. 9a and Supplementary Data 1), highlighting their contributions to the enhanced binding and potent agonism of MK-6892. This is consistent with our MD simulation analysis, which demonstrated a lower RMSD value for MK-6892 binding than for Niacin (Supplementary Fig. 9b).

## Unique allosteric agonist binding pocket

Compared to orthosteric ligands, allosteric modulators bind to non-conserved sites and offer promising opportunities for the development of novel therapeutic approaches with improved receptor selectivity and reduced side-effects[34–36]. However, despite their potential, only a limited number of class A GPCR structures in complex with positive allosteric modulators (PAMs) have been reported[37,38]; this highlights the necessity for further investigation into the binding and actions of PAMs. The existing structures reveal that PAMs bind to four broad regions within class A GPCRs (Fig. 4a). The crevice between TM3-ICL2-TM4 is the most frequently observed location, as evidenced by the structures of Cmpd-6FA-bound β2AR and LY3154207-bound DRD1 (Fig. 4a)[31,39]. Moreover, some PAMs bind superficially to the extracellular side of the receptor near the orthosteric pockets, such as LY2119620-bound M2R and ML382-bound MRGPRX1 (Fig. 4a). In addition, two unique positions located at the crevices between TM2-TM3-TM4 (ZCZ011-bound CB1R) and TM6-TM7-TM1 (MIPS521-bound A₁R) have also been identified as PAM binding sites (Fig. 4a)[40–43]. These structures provide valuable insights into the binding sites of PAMs and offer a foundation for the development of new therapeutic strategies.

As reported in previous studies[15,16], compound 9n ((R)−3-(4-iso-propylphenyl)−7-methyl-N-(1-phenoxypropan-2-yl)pyrazolo[1,5-a]pyr-imidine-6-carboxamide) is a partial allosteric agonist for HCAR2, with a potency of 1.0 μM (Fig. 4b). Moreover, it acts as a PAM of the orthosteric ligands Niacin and 3-HB, enhancing their potencies by ~19- and 12-fold, respectively (Fig. 4b, f and Supplementary Fig. 10b)[15,16]. To elucidate its binding site and mechanisms of action, we investigated two HCAR2 structures bound to compound 9n. The high-resolution structures revealed that compound 9n binds to a previously unidentified crevice between TM5-TM6-ECL2 on the extracellular side (Fig. 4c)[37,38]. Detailed analysis showed that compound 9n predominantly forms hydrophobic interactions with neighboring residues in this pocket (Fig. 4c). The phenoxy moiety of compound 9n inserts into a pocket formed by F186$^{5.36}$, H184$^{5.34}$, and L258$^{6.62}$; the pyr-azolopyrimidine moiety forms strong π-π interactions with F255$^{6.59}$;

and the isopropylphenyl moiety engages in extensive hydrophobic interactions with three hydrophobic residues on TM5, namely L194$^{5.44}$, L195$^{5.45}$, and F198$^{5.48}$ (Fig. 4c). In addition to these hydrophobic interactions, only one hydrogen bond is formed between the carbonyl group of compound 9n and the main chain of Q187$^{5.37}$ (Fig. 4c).

To link these structural findings with signaling outcomes, we carried out an exhaustive functional analysis of the residues implicated in compound 9n recognition (Fig. 4d, Supplementary Fig. 10a and Supplementary Data 1). Most alanine mutations in the pocket clearly reduced the PAM activities (Fig. 4d, Supplementary Fig. 10a and Supplementary Data 1). Remarkably, mutations of H184$^{5.34}$, F186$^{5.36}$ and L258$^{6.62}$ that involved in the recognition of phenoxy moiety significantly reduced the activities of compound 9n (Fig. 4d, Supplementary Fig. 10a and Supplementary Data 1), highlighting their critical roles and the importance of ECL2 in the actions of compound 9n. Sequence alignment of the residues involved in compound 9n recognition among HCARs revealed that HCAR3 maintains a similar interface, while HCAR1 is quite distinct (Fig. 4e). Our functional analysis is consistent with this finding and showed that compound 9n can serve as a PAM for HCAR3 with slightly reduced activity, but not for HCAR1 (Fig. 4f and Supplementary Fig. 10c). Recent studies have shown that the PAM activity of compound 9n operates by enhancing orthosteric ligand binding[15,16,44]. Indeed, our MD analysis also demonstrated that the frequency distribution of distances between both agonists and the critical residues within the orthosteric pocket (L280$^{7.39}$, Y284$^{7.43}$, and R111$^{3.36}$) is comparatively smaller in the presence of compound 9n binding (Supplementary Fig. 10d, e). Similar results were also observed in the more intuitive frequency distribution plots of ligand centroids (Supplementary Fig. 10f). Interestingly, compound 9n exhibits no PAM activity towards MK-6892-induced activation (Supplementary Fig. 10b, g). Consistently, structural alignment of the compound 9n-bound HCAR2 with the MK-6892-bound complex revealed that the outward movement of TM5 induced by MK-6892 could potentially lead to steric hindrance with compound 9n, thereby impeding the binding and action of compound 9n (Supplementary Fig. 10g).

## Discussion

Short-chain carboxylic acid molecules, such as lactate, butyrate, and 3-HB, play crucial physiological roles and are linked to various diseases[4,45]. Hydroxycarboxylic acid receptors, in turn, recognize these molecules, mediating a range of physiological functions[2,6]. Of particular interest, HCAR2 recognizes 3-HB and butyrate and represents a potential drug target for cardiovascular disease and neurogenic inflammatory diseases[7,19]. However, such drugs often elicit undesired flushing effects[18,46], highlighting the need for new active molecules and a deeper understanding of the ligand recognition mechanism of HCAR2.

Here, we present a comprehensive analysis of the HCAR2-Gil protein complex activated by diverse ligands, including endogenous 3-HB, essential human vitamin and prescription medication Niacin, potent synthetic orthosteric agonist, and allosteric agonist. Our results demonstrate that the conformation of HCAR2 is stabilized by three pairs of disulfide bonds located at the extracellular loops (Fig. 1c, d, Supplementary Fig. 6d and Supplementary Data 1), and this limits the size of the orthosteric pocket available for ligand recognition. Specifically, both 3-HB and Niacin, which are similar in size, bind to the pocket in a similar manner, interacting with TMs 2/3/7 and ECLs 1/2 of the receptor, with R111$^{3.36}$ playing a crucial role in recognizing the ligand carboxyl group (Fig. 2, Supplementary Fig. 7f and Supplementary Data 1). In contrast, the synthetic large ligand MK-6892 induces conformational changes in the receptor, leading to the formation of a larger ligand binding pocket that involves additional TM4 and TM5 (Fig. 3, Supplementary Fig. 9a and Supplementary Data 1). In addition, we investigated the binding mode of the allosteric ligand compound 9n, which acts as both an allosteric agonist and a PAM by enhancing endogenous ligand binding. Our

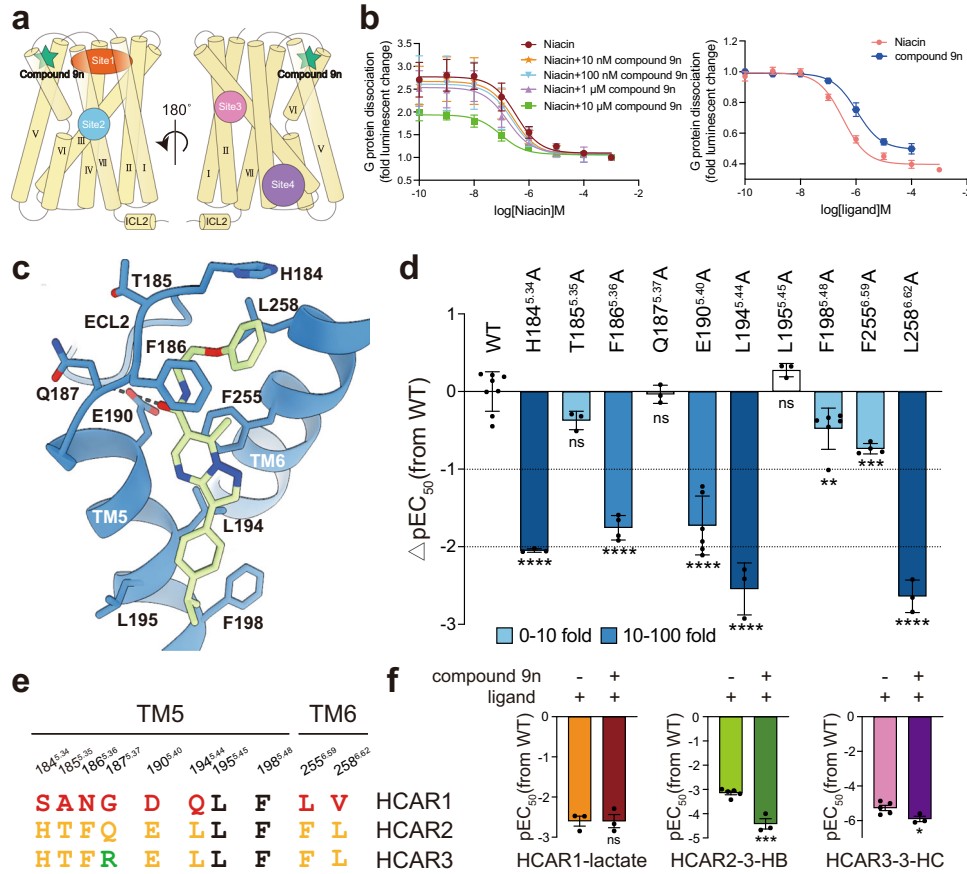

**Fig. 4 | Unique binding mode of the allosteric agonist compound 9n. a** The reported binding sites of PAMs in class A GPCRs. Site 1 (orange): LY2119620-bound M2R (PDB: 6OIK, iperoxo–LY2119620–M2R–Go), ML382-bound MRGPRX1 (PDB: 8DWG, BAM8-22–ML382-MRGPRX1-Gq), Site 2 (blue): MIPS521-bound A1R (PDB: 7LD3, adenosine–MIPS521–A1R–Gi2), Site 3 (pink): ZCZ011-bound CB1R (PDB: 7WV9, CP55940–ZCZ001–CB1-Gi), Cmpd-6FA-bound β2AR (PDB: 6N48, BI-167107–Cmpd-6FA–β2AR), Site 4 (purple): LY3154207-bound DRD1 (PDB: 7LJC, SKF-81297–LY3154207–DRD1–Gs). **b** Dose-response curves for the compound 9n-potentiated Niacin activation (left) and dose-response curves of Gi1 signaling induced by Niacin or compound 9n alone (right), error bars represent the standard deviation of curve fits from left ($n = 5, 4, 6, 5, 5$, and $6$ top to bottom) right ($n = 5, 6$ top to bottom) independent experiments. **c** Detailed interactions between compound 9n and HCAR2. Hydrogen bonds are depicted as black dashed lines. **d** Effects of mutations in the compound 9n binding pocket on its potentiated Niacin activation. The Gi1 dissociation signal was detected by NanoBiT assay in the presence of 25 nM Niacin at EC20 concentration and increasing concentrations of compound 9n. Bars represent differences in calculated agonist potency (pEC50) for each mutant relative to the wild-type receptor (WT). Data are colored according to the extent of effect. $^{ns}P > 0.05$; $^{*}P < 0.05$; $^{**}P < 0.01$; $^{***}P < 0.001$; $^{****}P < 0.0001$, one-way ANOVA followed by Dunnett's multiple comparison test, compared with the response of WT, (the detailed $P$ value for each condition is $P < 0.0001$, $P = 0.1983$, $P < 0.0001$, $P = 0.9997$, $P < 0.0001$, $P < 0.0001$, $P = 0.5425$, $P = 0.0066$, $P = 0.0001$, $P < 0.0001$, from left to right). Data are shown as the mean ± SEM from WT ($n = 8$), H184^5.34A ($n = 3$), T185^5.35A ($n = 3$), F186^5.36A ($n = 4$), Q187^5.37A ($n = 3$), E190^5.40A ($n = 6$), L194^5.44A ($n = 3$), L195^5.45A ($n = 3$), F198^5.48A ($n = 6$), F255^6.59A ($n = 4$), and L258^6.62A ($n = 3$) independent experiments. **e** Sequence alignment of compound 9n binding sites in the HCAR family. **f** PAM effect analysis of compound 9n in the HCAR family. Left to right: HCAR1: lactate; HCAR2: 3-HB; and HCAR3: 3-hydroxyoctanoic acid; $^{ns}P > 0.05$; $^{*}P < 0.05$; $^{***}P < 0.001$; $^{****}P < 0.0001$ (Data are the mean ± SEM from HCAR1-lactate ($n = 3, 3$), HCAR2-3-HB ($n = 5, 3$), and HCAR3-3-HC ($n = 5, 3$), independent experiments), Unpaired t test, two-tailed, with Welch's correction or Mann–Whitney test, compared with the response of WT, the detailed $P$ value for each condition is $P > 0.9999$, $P = 0.0005$, and $P = 0.0347$, from left to right). Source data are provided as a Source Data file.

results reveal that compound 9n binds to a novel allosteric pocket located at the outer side of the TMs 5/6 and ECL2 regions (Fig. 4, Supplementary Fig. 10a and Supplementary Data 1), providing unique insights into the allosteric modulation of GPCR. Recently, Yang et al. reported the structures of inactive and MK-6892-bound active HCAR2[47], providing important insights into the recognition and activation of synthetic agonizts to HCAR2. Structural comparison revealed that the binding modes of MK-6892 in the two structures are highly similar (Supplementary Fig. 9c). Our research significantly complements their study by offering a higher-resolution structural framework and invaluable insights into the binding and modes of action of diverse distinct ligands, which will accelerate the development of targeted drugs for HCAR2. However, our analysis of the recognition modes of the orthosteric agonist and compound 9n was based on the structures involving the simultaneous binding of both ligands. Despite their distinct binding pockets and absence of direct interactions, future investigations involving HCAR2 complexed with 3-HB, Niacin, and compound 9n alone will offer a more comprehensive understanding of the ligand recognition of HCAR2.

## Methods

### Constructs

The wild-type human HCAR2 was cloned into the pFastBac vector with hemagglutinin (HA) signal peptide and BRIL protein (the thermo-stabilized apocytochrome b562a) at the N-terminus. LgBit were cloned at the C-terminus followed by a double MBP tag and a TEV protease cleavage site between them[23]. A dominant-negative human Gαi1 (DNGαi1) was generated by site-directed mutagenesis to stabilize the interactions with the Gβγ subunits[20]. Human Gβ1 was cloned into the pFastBac dual vector together with Gγ2. In cellular signaling assays, the

wild-type receptor was sub-cloned into the pcDNA3.1 with the addition of an N-terminal Flag tag for cell-surface ELISA assay. The mutants were generated using site-directed mutagenesis. All the constructs were confirmed by sequencing. The primers used in this study are shown in Supplementary Data 3.

## Expression and purification of scFv16

The scFv16 was expressed and purified as previously described[47]. Briefly, the scFv16 with a 6×histidine tag was expressed in secreted form in *Trichoplusia ni* Hi5 insect cells for 48 h using the Bac-to-Bac system. The expressed scFv16 was purified using Ni-NTA resin, and the C-terminal 6xHis tag of the Ni-Nta eluent was cleaved by 3 °C protease. The protein was further purified by gel filtration chromatography using a Superdex 200 column. Finally, the purified scFv16 was concentrated and stored at −80 °C until further use.

## Complex formation and purification

HCAR2, DNGαi1 and Gβ1γ2 were co-expressed in Sf9 insect cells. The cells were cultured in ESF 921 serum-free medium (Expression Systems). Co-infection of the three types of baculoviruses was applied at a density of $2.4 \times 10^6$ cells per mL using a 1:1:1 ratio of HCAR2, DNGαi1, and Gβ1γ2. At 48 h post-infection, the cells were harvested and stored at −80 °C until subsequent use.

For purification, the cell pellets from 1 L of culture were thawed at room temperature and lysed using a buffer containing (in mM) 20 HEPES pH 7.5, 100 NaCl, and 2 $MgCl_2$ supplemented with EDTA-free protease inhibitor cocktail (Bimake) by Dounce homogenization. The HCAR2-Gi1 complexes were formed on membrane in the presence of agonist (100 μM Niacin + 100 μM compound 9n; 100 mM 3-HB + 100 μM compound 9n; 100 μM MK-6892), 1.0 mg scFv16, and apyrase (50 mU/ml, NEB) for 1 h. The membrane was subsequently solubilized with 0.5% (w/v) LMNG and 0.1% (w/v) CHS for 2.5 h at 4 °C and centrifuged at $30,000 \times g$ for 30 min. The resulting supernatant was incubated with MBP beads for 1 h at 4 °C. After that, the MBP beads were washed with five column volumes (CVs) of buffer containing 20 mM HEPES, pH 7.5, 100 mM NaCl, 2 mM $MgCl_2$, agonizts (10 μM Niacin +10 μM compound 9n; 10 mM 3-HB + 10 μM compound 9n; 10 μM MK-6892), 0.01% (w/v) LMNG and 0.005% (w/v) CHS. They were then eluted with 15 CVs of same buffer containing 10 mM maltose. The elute was collected and incubated with TEV protease for 1 h at room temperature, then concentrated with a 100 kDa cut-off concentrator (Millipore) and run on a Superose 6 Increase column. The eluted fractions were evaluated by SDS-PAGE, and those containing the receptor-Gi-protein complex were pooled and concentrated for cryo-EM experiments.

## Cryo-EM grid preparation and data collection

To prepare cryo-EM grids, 3.0 μl of the purified HCAR2–Gi1 complexes at ~15 mg/ml were applied onto glow-discharged holey carbon grids (Quantifoil, R1.2/1.3, 300 mesh). The grids were blotted for 3.5 s with a blot force of 10 at 4 °C, 100% humidity, and then plunge-frozen in liquid ethane using Vitrobot Mark IV (Thermo Fischer Scientific).

For 3-HB–bound and Niacin–bound HCAR2–Gi1 complexes, cryo-EM data were collected on a Titan Krios at 300 kV accelerating voltage at the Core Facilities, Zhejiang University Medical Center/Liangzhu laboratory. Micrographs were recorded using a Falcon 4 direct electron detector at a pixel size of 0.93 Å with EPU software. Image stacks were obtained at a dose rate of ~8.7 electrons per $Å^2$ per second with a defocus ranging from −1.0 to −2.0 μm. The total exposure time was 6 s, resulting in a total dose of 52 electrons per $Å^2$. A total of 5838 and 6893 movies were collected for the 3-HB– and Niacin–bound complexes, respectively.

For MK-6892–bound HCAR2–Gi1 complexes, data were automatically collected on a Titan Krios at 300 kV accelerating voltage in the Center of Cryo-Electron Microscopy (Zhejiang University). Micrographs were recorded using a Gatan K2 Summit Detector in super-resolution mode with a pixel size of 1.014 Å with SerialEM software[48]. Image stacks were obtained at a dose rate of about ~8.0 electrons per $Å^2$ per second with a defocus ranging from −1.0 to −2.0 μm. The total exposure time was 8 s, and 40 frames were recorded per micrograph. A total of 2948 movies were collected for the MK-6892–bound complex.

## Cryo-EM data processing

For the 3-HB–bound HCAR2-Gi1 complex, image stacks were aligned with RELION 4.0[49]. Contrast transfer function (CTF) parameters were estimated by Gctf v1.18[50]. Automated particle selection using Topaz picking in RELION produced 4,793,474 particles. A map of the SSTR2–Gi1 complex (EMD-32528) low-pass filtered to 60 Å was used as the initial reference map[51]. The particles were imported to CryoSPARC v3.32 for 2D classification[52], Ab-initio Reconstruction and 2 rounds of heterogeneous refinement to discard fuzzy particles, resulting in 2,514,582 particles. A further 2 rounds of local 3D classification on the receptor in RELION, generated one good subset with 157,251 particles. The final good particles were subjected to 3D refinement, CTF refinement, and Bayesian polishing. The final refinement generated a map with an indicated global resolution of 2.60 Å at a Fourier shell correlation of 0.143. The final map was sharpened with RELION and used for subsequent model-building and analysis.

For the Niacin–bound complex, image stacks were aligned with RELION 4.0. CTF parameters were estimated by Gctf v1.18. A total of 5,934,873 particles generated from the template-based particle picking were subjected to 2 rounds of 3D classification in RELION. A further 2 rounds of local 3D classifications focusing the alignment on the receptor produced one high-quality subset accounting for 321,148 particles, which were subsequently subjected to 3D refinement, CTF refinement, and Bayesian polishing. The final refinement generated a map with an indicated global resolution of 2.55 Å at a Fourier shell correlation of 0.143. The final map was sharpened with RELION and used for subsequent model-building and analysis.

For the MK-6892–bound complex, cryo-EM movies were aligned using MotionCor 2.1 and CTF parameters were estimated by Gctf v1.18. Template-based particle-picking in RELION produced 2,467,658 particles, which were subjected to 2 rounds of 3D classification in RELION. A further 3 rounds of local 3D classification on the receptor produced one high-quality subset accounting for 413,723 particles, which were subsequently subjected to 3D refinement, CTF refinement and Bayesian polishing. The final refinement generated a map with an indicated global resolution of 2.76 Å. The final map was sharpened with deepEMhancer[53] and used for subsequent model building and analysis.

## Model building and refinement

The AlphaFold-predicted structure of HCAR2 was used to generate an initial model of the receptor[54]. The atomic coordinates of Gi1 and scFv16 from the structure of the SSTR2–Gi1 complex (PDB: 7WIC) (SS-14–SSTR2–Gi1)[51] were used to generate an initial model of the Gi1–scFv16 complex. Models were manually docked into the density maps using UCSF Chimera. Agonist coordinates and geometric restraints were generated using a phenix.elbow[55]. The initial models were subjected to flexible fitting using Rosetta and were further rebuilt in Coot and real-space-refined in Phenix. The final refinement statistics were validated using the module 'comprehensive validation (cryo-EM)' in Phenix[55]. The goodness-of-fit of the models to the maps were determined using a global model-versus-map Fourier shell correlation. The refinement statistics are provided in Supplementary Table S1. Structural figures were created using the UCSF Chimera X package[56].

## NanoBiT G-protein dissociation assay

The Gi1-protein recruitment assay was applied using the NanoBiT system (Promega) as previously described[26,57]. The C-terminus of the HCAR2 was fused to LgBiT of the NanoBiT luciferase via 15-amino-acid flexible linkers. The Gβ1 subunit was C-terminally fused to SmBiT via a

15-amino-acid flexible linker. Human HCAR2-LgBit, WT Gαi1, SmBiT-fused Gβ1 and Gγ2 were co-expressed in CHO-K1 cells (ATCC, CCL-61) using transfection. Six hours after transfection, the cells were cultured in 96-well plates. Forty-eight hours later, the cells were washed with HBSS and loaded with 40 μL Coelenterazine. The baseline signals were immediately read for five cycles after a 30-minute incubation. Following agonist addition, the signals were read for an additional 30 cycles. Luminescence counts were normalized to the initial count, and fold-change signals over vehicle treatment were used to demonstrate the G-protein-binding response. The primers used in this study are shown in Supplementary Data 3.

### Cell-surface ELISA

After transfection of HCAR2 or mutants for 24 h, cells were seeded into 96-well culture plates and incubated for another 24 h at 37 °C under 5% $CO_2$. Cells were fixed in 4% polyformaldehyde for 10 min and blocked with 1% (w/v) BSA for 1 h at room temperature. The ELISA plates were washed three times with phosphate-buffered saline (PBS) and incubated with the monoclonal anti-FLAG HRP conjugate (Sigma Aldrich, Cat F1804). After 30 min, the cells were washed three times with PBS, and then chemiluminescence was assessed by the addition of 50 μL of HRP substrate (Thermo Fisher, Cat 37069)[27]. Chemiluminescence values were normalized to wild-type receptor and graphed as a percentage of wild-type using Graphpad Prism 9 (Graphpad Software Inc., San Diego, CA). Also see Supplementary Data 2 for raw data relating to the cell surface expression reported in Figs. 2–4.

### Molecular dynamics simulations

To investigate the stability of different agonizts in the binding pocket of HCAR2, we generated MD simulations of Niacin-bound, 3-HB-bound, and MK-6892-bound HCAR2 models, which were derived from the structures of the Niacin-HCAR2-Gi1, 3-HB-HCAR2-Gi1, and MK-6892-HCAR2-Gi1 complexes, respectively, that were constructed from our cryo-EM map data. In addition, to assess the effect of compound 9n in the above systems, extra MD simulations for the Niacin-bound and 3-HB-bound HCAR2 models were also generated with the presence of compound 9n. Three and six individual MD simulation trajectories were generated for the systems with and without the compound 9n, respectively (Supplememtary Data 4 and 5. The receptor orientations were obtained from the Orientations of Proteins in Membranes database and all input files were generated using the CHARMM-GUI website[58,59]. These complexes were embedded in an asymmetric lipid bilayer representing the plasma membrane[60]. The outer leaflet was composed of POPC (33.3 mol%), PSM (33.3 mol%), and cholesterol (33.3 mol%); while the inner leaflet was composed of POPC (35 mol%), POPE (25 mol%), POPS (20 mol%), and cholesterol (20 mol%). The bilayer membrane was finally generated with a total number of 200 lipids. All systems were solvated in a rectangular box with dimensions of $8.4 \times 8.4 \times 11.2$ nm³, and filled up with the TIP3P water. To balance the charge of the complex system, 0.15 M NaCl was added. All simulations were generated with GROMACS 2021.5[61,62] using the CHARMM 36-mm Force Field[63] for all solutes. These complex systems were processed for energy minimization for 5000 steps using the steepest descent method, and a 125 ps NVT simulation at 310.15 K was generated for the solvent equilibration using the Berendsen thermostat with heavy atoms restrained at 10.0 kcal mol⁻¹ Å⁻². The equilibration was processed for an additional five cycles, during which the harmonic restraints were 5.0, 2.5, 1.0, 0.5, and 0.1 kcal mol⁻¹ Å⁻². The equilibration was applied at 310.15 K and 1 atm for 1 ns in NPT ensemble, using the Berendsen thermostat and barostat. Finally, the production simulations were run at 310.15 K and 1 atm in the NPT ensemble for 1000 ns with a time step of 2 fs, using the Nosé–Hoover Langevin thermostat and the Parrinello-Rahman barostat. A cutoff of 12 Å was used for the van der Waals and short-range electrostatic interactions. Long-range electrostatic interactions were treated by the Particle mesh Ewald

algorithm[64]. Force-based switching was applied for soft changing of interactions over 10–12 Å. The covalent bonds containing hydrogen atoms were constrained using the LINCS algorithm. All analyses were completed in Gromacs 2021.5, MDTraj v1.9.8[65] and VMD[66]. The system setup for MD simulations, configuration used for MD production simulations and all initial and final coordinates of MD simulations are provided in Supplementary Data 4 and 5.

### Molecular docking

In order to verify the accuracy of agonizts (Niacin and 3-HB), we performed molecular docking of these compounds to HCAR2. The structures of Niacin, 3-HB and HCAR2 were derived from the models of Niacin-HCAR2-Gi1 and 3-HB-HCAR2-Gi1 complexes, respectively. These agonists and receptors were then converted to PDBQT format using the rdkit2pdbqt.py script from watvina. Docking was conducted using AutoDock Vina v.1.1.2 and the docking box was set using AutoDockTools-1.5.6 software[67]. To improve accuracy, the parameter exhaustiveness was set to 80 with 5 postures to generate for each agonist. Finally, we selected the top three binding modes for each compound, which closely resembled our model, and visualized these modes using ChimeraX v.1.2.5[68].

### Statistical analysis

All data are presented as the mean ± SEM from at least three independent experiments. Statistical analyses were applied using Graph-Pad Prism 9 software. For data that passed normality and equal variance tests, comparisons between two groups were measured by Student's $t$ test, differences among three or more groups were examined by one-way analysis of variance (ANOVA) followed by Dunnett's multiple comparison test or Tukey's multiple comparison test. For dose–response experiments, data were normalized and analyzed using nonlinear curve fitting for the log (agonist) versus response (three parameters) curves.

### Reporting summary

Further information on research design is available in the Nature Portfolio Reporting Summary linked to this article.

## Data availability

The atomic coordinates and the electron microscopy maps of the 3-HB–compound 9n–HCAR2–Gi1 complex, Niacin–compound 9n–HCAR2–Gi1 complex and MK-6892-HCAR2–Gi1 complex have been deposited in the Protein Data Bank (PDB) under accession numbers 8J6Q, 8J6P and 8J6R, and the Electron Microscopy Data Bank (EMDB) under accession codes EMD-36011, EMD-36010 and EMD-36012, respectively. All data analyzed in this study are included in this paper and its Supplementary Information. The structural data used in this study were retrieved from the PDB using accession codes 6OIK (iperoxo–LY2119620–M2R-Go) 8DWG (BAM8-22-ML382-MRGPRX1-Gq) 7LD3 (adenosine–MIPS521–A₁R–Gi2), 7WV9 (CP55940–ZCZ001–CB1-Gi), 6N48 (BI-167107–Cmpd-6FA–β2AR), 7LJC (SKF-81297–LY3154207–DRD1-Gs), 7WIC (SS-14–SSTR2-Gi1), 7DFL (histamine-H1R-Gq), 7LJD (dopamine–DRD1-Gs), 7E2Y (5-HT–5-HT1A-Gi). Source data are provided with this paper.

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

## Acknowledgements

The cryo-EM data were collected at the Cryo-Electron Microscopy Facility, Zhejiang University Medical Center/Liangzhu laboratory and the Cryo-Electron Microscopy Center, Zhejiang University. Protein purification was performed at the Protein Facilities, Zhejiang University School of Medicine. This project was supported by the Zhejiang Province Natural Science Fund for Excellent Young Scholars (LR22C050002 to C.M.); the Ministry of Science and Technology (2019YFA050880 to Y.Z); the National Natural Science Foundation of China (82025005 to X.M.; 32100959 to C.M.; 82322070 to C.M.; 32371249 to C.M.); the Key R&D Projects of Zhejiang Province (2021C03039 to Y.Z.); the Fundamental Research Funds for the Central Universities (2019XZZX001-01-06 to Y.Z.). We thank Prof. Iain C. Bruce (University of Hong Kong, China) for reviewing the manuscript.

## Author contributions

Y.Z., X.M., and C.M. conceived and supervised the whole project; C.M. initiated the project and purified the MK-6892-bound HCAR2-Gi1 complex; M.G. purified the 3-HB and Niacin-bound HCAR2-Gi1 complexes; D.-D.S. evaluated the sample by negative-stain EM; L.-N.C. prepared the cryo-EM grids; S.-K.Z. collected the cryo-EM data; C.M. and S.-K.Z. performed cryo-EM map calculation and model building; M.G. and Y.Zhu. generated the constructs of mutants and performed the cellular functional assays; S.-K.Z. performed molecular dynamics simulation and docking analysis; C.M., S.-K.Z. and M.G. performed structural analysis. M.G., S.-K.Z. and Y.Zhu. prepared the figures; Y.L., L.Y., Z.W., H.Z., W.-W.W. and Q.S. participated in data analysis; C.M. prepared the draft of the manuscript; Y.L. edited the manuscript; Y.Z., X.M., and C.M. wrote the manuscript with input from all authors.

## Competing interests

The authors declare no competing interests.
