## [Peer Review File · Nature Communications]

REVIEWER COMMENTS

Reviewer #1 (Remarks to the Author):

This is a very good paper showing three cryo-EM structures of hydroxycarboxylic acid receptor HCAR2, a member of GPCR family, with its Gi protein and three agonists. Additionally, an allosteric agonist, positive allosteric modulator (compound 9n), is found in the structures of orthosteric agonist 3-HB and niacin. The methodology, involving structural, functional and computational analyses, is valid and the obtained results are clear and convincing. They are also compared to the recent paper of Yang et al. [Nat. Commun. 2023] which shows structure of inactive HCAR2 as well as HCAR2-Gi complex with the synthetic agonist MK-6892.

1. Since the agonist MK-6892 in complex with HCAR2 is the same in both papers it would be good to superimpose both structures and compare similarities and differences in the binding modes. This can be shown in the supplementary material.
2. The MD simulations were conducted for agonists to show their stability in the binding site. It was found that 3-HB is escaping from the binding site, however, no escaping path(s) is shown. It is interesting to show where this compound escapes in some simulations.
3. Fig. 4c shows the allosteric compound 9n, however, there is MK-6892 in the figure legend. Since 9n is a PAM it affects agonist binding. To reveal allosteric-orthosteric coupling, one can conduct MD simulations of HCAR2 with both compounds, agonist and PAM, and compare to MD simulations with agonist alone. It could be done via dynamic fingerprints or other type of analysis.
4. Methods: percentage of particular lipids in the membrane is specified but the total number of lipids is missing. There are no dimensions of periodic box.
5. Methods: "The particle mesh Ewald method was used to calculate electrostatic interactions with a cut-off of 12 Å" is imprecise. What was the cutoff for van der Waals and short-range electrostatic interactions and whether the switching function range was used? There is no reference to PME method and no parameters are specified for this method.
6. Methods: "The hydrogen bonds were constraint using the LINCS algorithm" is incorrect. Probably it should be: "The covalent bonds containing hydrogen atoms were constrained using the LINCS algorithm".

Reviewer #2 (Remarks to the Author):

In the present paper Mao and coworkers present three novel cryo-EM structures of the important HCAR2 receptor in complex with 1) the endogenous 3-OH butyrate (3-HB) plus an

allosteric agonist/allosteric modulator 9n; 2) the classical agonist niacin plus 9n; and 3) the potent synthetic agonist MK-6892. They complement this structural biology work with a rather comprehensive mutational and molecular pharmacological analysis of the binding pockets. The paper is well written with a clear well-balanced narrative and generally well-illustrated although with space for improvement (see below).

Although a recent paper also in Nature Comm and one in BioRxiv have just presented HCAR2 in complex with MK-6892, the present paper provides truly novel observations and information concerning e.g. the binding of the endogenous ligand, 3-HB as well as the allosteric agonist 9n – and is particularly well written and with a better balance of the literature, except for the story about dyslipidemia, which is presented wrongly in all three papers (see point #1).

Major points:

1. HCAR2 and dyslipidemia – It is wrongly stated in the abstract, the introduction, and in the discussion that HCAR2 is a target for drugs against dyslipidemia. It was however shown almost a decade ago that niacin does NOT mediate its beneficial effect on dyslipidemia through HCAR2, but through an unknown target. Anti-inflammatory effects of nicotinic acid in human monocytes are mediated by GPR109A dependent mechanisms (Digby JE et al: *Arterioscler Thromb Vasc Biol.* 2012; 32:669-76). In contrast only the flushing side effect is mediated through HCAR2. The authors of the present paper does however tell that HCAR2 is a potential target for anti-inflammatory drugs targeting in particular neurogenic inflammation and e.g. MS. Please justify your interesting structural biology story by focusing on that - and get rid of the wrong dyslipidemia story.
2. In lines 89 -92 the three structures are identified. However in the rest of the paper the binding sites of 3-BH, niacin, and 9n (and MK-6892) are discussed as if they were in separate structures. The fact that the binding sites for 3-BH and niacin both must be influenced by the fact that they are bound not alone but together with 9n should be clearly illustrated and discussed (Figure presenting the two dual binding modes – binding of 9n relative to 3-BH and niacin). Also please illustrate the relative binding of 9n to MK-6892.
3. The authors very well present how the agonist binding sites in HCAR2 are totally closed off from the exterior by the disulfide-stabilized cap or lid. This opens for the question of how the agonists get to their binding sites? Does the lid open up? Or more likely, where do the agonist enter between the exterior segments of the TMs? One very interesting potential possibility is that the agonists enter between TM-V and TM-VI, i.e. at the site where 9n binds. If this is the case 9n could work in a similar manner as some muscarinic allosteric modulators which 'locks the door' after the agonist has bound? (See next point) The unbinding could be directly studied by MD simulations.
4. MD simulations – It is not clear for the reader what actually happens during your MD simulations. Firstly, please tell whether/ that you do the MD simulations after removing the allosteric agonist 9n from the structures? Secondly, please do not only use RMSD for the ligand as it is unclear whether the ligand actually leaves the binding site? It could just move around in the site, which probably is more 'loose after the likely removal of 9n? Please indicate changes in some distance from the ligand to e.g. R111.
5. (minor point) Molecular pharmacology of 9n – 9n does not really change the dose-response curve for niacin unless added at 10 μ M (10⁻⁵) (Fig. 4b), which according to the dose-response

curve for 9n alone should have given full agonism. Nevertheless this concentration of 9n 'only' provides 50% agonism? Why?

Reviewer #3 (Remarks to the Author):

Mao et al. presenta manuscript describing the binding mode of endogenous and synthetic ligands to HCAR2, a receptor involved in the metabolism of hydrocarboxylic acids and a target for the treatment of dyslipidemia. The authors present three cryo-EM structures of HCAR2 coupled to a Gi hetrotrimer and bound to the endogenous 3-HB, niacin and a synthetic MK-6892, where an allosteric compound (compound 9n) is additionally present at the 3-HB and niacin structures. The authors show functional and computational data supporting the binding modes seen in the structural data. The manuscript is well-written and organized, and present relevant data. Although, the recent publication of HCAR2 structures has an impact on the novelty of the work, the authors present a significant amount of additional data and functional assays.

In my opinion the manuscript is recommended for publication with minor modifications:

1.- The challenge of studying such small agonists is being certain about the binding pose of such ligands within the cryo-EM maps, which are generally of lower resolution when compared to x-ray structures. In this case, the authors manage to achieve relatively high resolution and support the binding mode of 3-HB and niacin with docking and MD simulations. However the authors should acknowledge that, based on experimental maps, niacin and 3-HB could be fitted the opposite orientation within the map. Alternatively, the authors should elaborate slightly more to defend the choosen orientation of both ligands.

2.- The authors comment separately on the binding modes on 3-HB and niacin and compound9, however the structures of 3-HB and niacin in the absence of compound9 have not been obtained, hence the authors should include a comment about the potential influence on the allosteric compound9 on the binding mode of 3-HB and niacin.

Responses to the reviewers' comments

We thank the referees for their valuable time reviewing our manuscript and the constructive suggestions that they have provided. We have carefully taken these comments into consideration when preparing a revision, which resulted in a more thorough and clear manuscript. We have copied each comment in **Black Italic**, which is followed by our own point-by-point response in **Blue**, including details about the corresponding changes to the manuscript.

Reviewer #1

This is a very good paper showing three cryo-EM structures of hydroxycarboxylic acid receptor HCAR2, a member of GPCR family, with its Gi protein and three agonists. Additionally, an allosteric agonist, positive allosteric modulator (compound 9n), is found in the structures of orthosteric agonist 3-HB and Niacin. The methodology, involving structural, functional and computational analyses, is valid and the obtained results are clear and convincing. They are also compared to the recent paper of Yang et al. [Nat. Commun. 2023] which shows structure of inactive HCAR2 as well as HCAR2-Gi complex with the synthetic agonist MK-6892.

Response: We thank the reviewer for the positive comments on our manuscript.

1. Since the agonist MK-6892 in complex with HCAR2 is the same in both papers it would be good to superimpose both structures and compare similarities and differences in the binding modes. This can be shown in the supplementary material.

Response: We thank the referee for his/her insightful suggestions. Structural comparison between our resolved MK-6892-bound HCAR2 and the recently reported structure (PDB 7XK2) reveals that the binding mode of MK-6892 in the two structures are highly similar. However, our higher-resolution structure offers a more precise depiction of the ligand binding interactions. First, our structure shows a 2.87Å movement of the side-chain of Y284^{7,43} towards the agonist, establishing a hydrogen bond with the carboxyl group of MK-6892. Second, compared to recent structure, our structure demonstrates a rotation of approximately 90 degrees in the hydroxypyridine moiety, resulting in hydrophobic and hydrogen-bonding interactions with H161^{4,59} (Supplementary Fig.9c). We have added these figures and descriptions in our revised manuscript (Supplementary Fig.9c and Lines 307-309).

Supplementary Fig.9 | Recognition of MK-6892 by HCAR2.

c Structural comparison of our resolved MK-6892-bound HCAR2 (red) with the recently reported structure (PDB 7XK2; pink). Our higher-resolution structure offers a

more precise depiction of the ligand binding interactions. First, our structure shows a 2.9 Å movement of the side-chain of Y284^{7,43} towards the agonist, establishing a hydrogen bond with the carboxyl group of MK-6892. Second, compared to recent structure, our structure demonstrates a rotation of approximately 90 degrees in the hydroxypyridine moiety of MK-6892, resulting in hydrophobic and hydrogen-bonding interactions with H161^{4,59}.

2. *The MD simulations were conducted for agonists to show their stability in the binding site. It was found that 3-HB is escaping from the binding site, however, no escaping path(s) is shown. It is interesting to show where this compound escapes in some simulations.*

Response: We thank the referee for his/her insightful suggestions. Our MD simulations indicated that the N-terminal and ECL2 are relatively flexible, leading to the potential escape of the unstable 3-HB from the crevice formed by the TM4-TM5-ECL2 region (Supplementary Fig.7d). We have added these descriptions and figures in our revised manuscript (Supplementary Fig.7d and Line 150).

Supplementary Fig.7 | Recognition of 3-HB and Niacin by HCAR2.

d The potential escaping pathway of 3-HB revealed by MD simulation. As illustrated, the ECL2 exhibits an upward shift from its initial state (deep green) to an intermediate state (gray), thereby creating a crevice by the TM4-TM5-ECL2 region. Within this spatial context, 3-HB (orange) define the plausible pathway for its escape.

3. *Fig. 4c shows the allosteric compound 9n, however, there is MK-6892 in the figure legend. Since 9n is a PAM it affects agonist binding. To reveal allosteric-orthosteric coupling, one can conduct MD simulations of HCAR2 with both compounds, agonist and PAM, and compare to MD simulations with agonist alone. It could be done via dynamic fingerprints or other type of analysis.*

Response: We thank the referee for pointing out the problem. We have corrected the MK-6892 to compound 9n in the figure legend of Fig. 4c (Line 781). To elucidate allosteric-orthosteric coupling, we undertook MD simulations involving 3-HB and Niacin-bound HCAR2, both with and without the presence of compound 9n. We analyzed the frequency distribution plots of ligand centroids. It is clear that Niacin demonstrates a more confined distribution in the presence of compound 9n. As for 3-HB, its localization within the center of the pocket is notably more probable in the presence of compound 9n (Supplementary Fig.10f). These results suggest that compound 9n plays a role in stabilizing the ligand binding. We have added the corresponding description and figures in the revised manuscript (Supplementary

Fig.10f and Lines 271-272).

Supplementary Fig.10 | Recognition of compound 9n by HCAR2.

f Frequency distribution plot of agonists centroids with or without compound 9n (light green). Frequency distribution is colored by spectrum from high (red) to low (blue), initial positions of agonists are shown in magenta.

4. *Methods: percentage of particular lipids in the membrane is specified but the total number of lipids is missing. There are no dimensions of periodic box.*

Response: We thank the referee for the careful reviewing. The bilayer membrane was finally generated with a total number of 200 lipids. All systems were solvated in a rectangular box with dimensions of $8.4 \times 8.4 \times 11.2 \text{ nm}^3$, and filled up with the TIP3P water. We have added the descriptions in our revised manuscript (Lines 498-503 in Methods).

5. *Methods: “The particle mesh Ewald method was used to calculate electrostatic interactions with a cut-off of 12 \AA ” is imprecise. What was the cutoff for van der Waals and short-range electrostatic interactions and whether the switching function range was used? There is no reference to PME method and no parameters are specified for this method.*

Response: We thank the referee for the careful reviewing. A cutoff of 12 \AA was used for the van der Waals and short-range electrostatic interactions. Long-range electrostatic interactions were treated by the Particle mesh Ewald algorithm². Force-based switching was applied for soft changing of interactions over 10 to 12 \AA . We have added the descriptions and reference in our revised manuscript (Lines 509-512 in Methods).

6. *Methods: “The hydrogen bonds were constraint using the LINCS algorithm” is incorrect. Probably it should be: “The covalent bonds containing hydrogen atoms were*

constrained using the LINCS algorithm”.

Response: We thank the referee for pointing out the problem. We have revised it accordingly (Lines 512-513 in Methods).

Reviewer #2

In the present paper Mao and coworkers present three novel cryo-EM structures of the important HCAR2 receptor in complex with 1) the endogenous 3-OH butyrate (3-HB) plus an allosteric agonist/allosteric modulator 9n; 2) the classical agonist Niacin plus 9n; and 3) the potent synthetic agonist MK-6892. They complement this structural biology work with a rather comprehensive mutational and molecular pharmacological analysis of the binding pockets. Although a recent paper also in Nature Comm and one in BioRxiv have just presented HCAR2 in complex with MK-6892, the present paper provides truly novel observations and information concerning e.g. the binding of the endogenous ligand, 3-HB as well as the allosteric agonist 9n – and is particularly well written and with a better balance of the literature, except for the story about dyslipidemia, which is presented wrongly in all three papers (see point #1).

Response: We are grateful to the reviewer’s positive assessment on the novelty and quality of our study.

Major points:

1. HCAR2 and dyslipidemia – It is wrongly stated in the abstract, the introduction, and in the discussion that HCAR2 is a target for drugs against dyslipidemia. It was however shown almost a decade ago that Niacin does NOT mediate its beneficial effect on dyslipidemia through HCAR2, but through an unknown target. Anti-inflammatory effects of nicotinic acid in human monocytes are mediated by GPR109A dependent mechanisms (Digby JE et al: Arterioscler Thromb Vasc Biol. 2012; 32:669-76). In contrast only the flushing side effect is mediated through HCAR2. The authors of the present paper does however tell that HCAR2 is a potential target for anti-inflammatory drugs targeting in particular neurogenic inflammation and e.g. MS. Please justify your interesting structural biology story by focusing on that - and get rid of the wrong dyslipidemia story.

Response: We appreciate the referee's insightful comments and suggestions. We agree with the referee's viewpoint regarding the intricate and contentious nature of the relationship between HCAR2 and the lipid-lowering activity of niacin. To avoid the potential misleading, we have revised the descriptions in the manuscript as follows:

Lines 27-30: We have revised the sentences as follows: “HCAR2 is one such receptor, activated by endogenous β -hydroxybutyrate (3-HB) and butyrate, as well as targeted by Niacin, which is under intensive study due to its implications in cardiovascular and neuroinflammatory diseases.”

Lines 59-62: We have revised the sentences as follows: “Importantly, HCAR2 stands as a potential therapeutic target for neuroimmune disorders, cardiovascular diseases and cancers. However, the pursuit of drug discovery targeting HCAR2 is frequently hampered by common adverse effects, such as the flushing.”

Lines 283-285: We have revised the sentences as follows: “Of particular interest, HCAR2 recognizes 3-HB and butyrate and represents a potential drug target for cardiovascular disease, neurogenic inflammatory diseases and multiple sclerosis.”

2. In lines 89 -92 the three structures are identified. However in the rest of the paper the binding sites of 3-HB, Niacin, and 9n (and MK-6892) are discussed as if they were

in separate structures. The fact that the binding sites for 3-BH and Niacin both must be influenced by the fact that they are bound not alone but together with 9n should be clearly illustrated and discussed (Figure presenting the two dual binding modes – binding of 9n relative to 3-BH and Niacin). Also please illustrate the relative binding of 9n to MK-6892.

Response: We thank the referee's valuable suggestion. We have added the corresponding descriptions detailing the binding interactions of 3-HB and Niacin in the presence of compound 9n to the legends of all relevant figures in the revised manuscript (Fig.2 and Lines 746-747 Fig.3 and Line 758, supplementary material Lines 9, 12, 17, 20, 34, 39, 80). Furthermore, to avoid the potential misleading, we have revised the manuscript to include the following statement in the discussion section: "Our analysis of the recognition modes of the orthosteric agonist and compound 9n is based on the structures involving simultaneous binding of both ligands. Despite their distinct binding pockets and absence of direct interactions, future investigations involving HCAR2 complexed with 3-HB, Niacin, and compound 9n alone will offer a more comprehensive understanding of the ligand recognition of HCAR2." (Line 312-317) For the relative binding of compound 9n to MK-6892, our functional assay showed that compound 9n exhibits no PAM activity to MK-6892 activation(Supplementary Fig.10b). Consistently, structural alignment of the compound 9n-bound HCAR2 with the MK-6892-bound complex showed that MK-6892-induced outward movement of TM5 could indeed create steric hindrance with compound 9n (Supplementary Fig.10g). We have added the corresponding description and figures in the revised manuscript (Supplementary Fig.10b and g, and Lines 272-277)

Supplementary Fig.10 | Recognition of compound 9n by HCAR2.

b Dose-response curves for the compound 9n-potentiated MK-6892 activation.

Supplementary Fig.10 | Recognition of compound 9n by HCAR2.

g Structural superimposition of the Niacin and compound 9n (light green) -bound HCAR2 (blue) with the MK-6892-bound complex (red) showed that MK-6892-induced outward movement of TM5 would create steric hindrance with compound 9n.

3. The authors very well present how the agonist binding sites in HCAR2 are totally closed off from the exterior by the disulfide-stabilized cap or lid. This opens for the question of how the agonists get to their binding sites? Does the lid open up? Or more likely, where do the agonists enter between the exterior segments of the TMs? One very interesting potential possibility is that the agonists enter between TM-V and TM-VI, i.e. at the site where 9n binds. If this is the case 9n could work in a similar manner as some muscarinic allosteric modulators which ‘locks the door’ after the agonist has bound? (See next point) The unbinding could be directly studied by MD simulations.

Response: We thank the referee’s valuable comments. In our MD simulations involving 3-HB-bound HCAR2, we observed the escape of 3-HB from the ligand pocket in four out of six replicates (Supplementary Fig.9b and Line 150). By analyzing the trajectory files, we found that the N-terminal and ECL2 are relative flexible, thereby enabling 3-HB to exit through the crevice formed by the TM4-TM5-ECL2 region (Supplementary Fig.7d and Line 150). This observation leads us to speculate that the potential escape route for both 3-HB and Niacin is via the crevice created by TM4-TM5-ECL2 (Supplementary Fig.7d). Regarding compound 9n, our MD simulations, as well as the previous study, indicated that its PAM effect primarily results from its ability to stabilize receptor conformation and enhance agonist binding. We will discuss it in detail in the next point.

Supplementary Fig.7 | Recognition of 3-HB and Niacin by HCAR2.

d The potential escaping pathway of 3-HB revealed by MD simulation. As illustrated, the ECL2 exhibits an upward shift from its initial state (deep green) to an intermediate state (gray), thereby creating a crevice by the TM4-TM5-ECL2 region. Within this spatial context, 3-HB (orange) define the plausible pathway for its escape.

4. MD simulations – It is not clear for the reader what actually happens during your MD simulations. Firstly, please tell whether/ that you do the MD simulations after removing the allosteric agonist 9n from the structures? Secondly, please do not only use RMSD for the ligand as it is unclear whether the ligand actually leaves the binding site? It could just move around in the site, which probably is more ‘loose after the likely removal of 9n? Please indicate changes in some distance from the ligand to e.g. R111.

Response: We appreciate the referee's insightful comments and suggestions. Initially, our MD simulations involving 3-HB and Niacin were carried out in the presence of compound 9n. To get a comprehensive understanding of the action of these ligands, we subsequently performed additional MD simulations focusing on the agonists alone, without the presence of compound 9n. We also acknowledge the value of the referee's suggestion. As such, we conducted an analysis involving the frequency distribution of distances between the agonists and the key residues L280^{7,39}, Y284^{7,43}, and R111^{3,36} for all simulations (Supplementary Fig.10d and e). The outcomes of this analysis consistently revealed that, on average, the distances between the two agonists and these three crucial residues within the orthosteric pocket were clearly smaller in the presence of compound 9n binding. This observation is consistent with prior research suggesting that compound 9n facilitates the binding of orthosteric ligands. We have included corresponding descriptions and Figures within our revised manuscript (Supplementary Fig.10d and Lines 267-271).

Supplementary Fig.10 | Recognition of compound 9n by HCAR2.

d Frequency distribution of distances between Niacin and L280^{7,39}, Y284^{7,43}, and R111^{3,36} in the presence (light green) and absence (cyan) of compound 9n, respectively. **e** Frequency distribution of distances between 3-HB and L280^{7,39}, Y284^{7,43}, and R111^{3,36} in the presence (light green) and absence (lavender) of compound 9n, respectively.

5. (minor point) Molecular pharmacology of 9n – 9n does not really change the dose-response curve for Niacin unless added at 10 μM (10^{-5}) (Fig. 4b), which according to the dose-response curve for 9n alone should have given full agonism. Nevertheless this concentration of 9n ‘only’ provides 50% agonism? Why?

Response: We thank the referee for his/her valuable comments. It is indeed noteworthy that the dose-response curves depicting the potentiated Niacin activation by compound 9n exhibit a milder effect at lower concentrations of compound 9n, which is consistent with the previous study⁵. This phenomenon may result from the combination of the modest PAM effect of compound 9n and the relative strong agonism of Niacin. Consistently, with regard to the weak agonist 3-HB, our data illustrate a more pronounced impact of compound 9n on the dose-response curves for 3-HB in contrast to Niacin (Supplementary Fig.10b).

Regarding the agonism of compound 9n, we thank the referee for pointing out the problem. Both our study and a previous investigation showed that compound 9n could function as a partial agonist^{15,16}. In Fig. 4b, for clarity, all dose-response curves have been normalized to the final luminescence readout. To avoid the misleading, we have revised Fig. 4b and relocated the dose-response curves of compound 9n to Fig. 4b left and right.

Fig.4 | Unique binding mode of the allosteric agonist compound 9n.

b Dose-response curves for the compound 9n-potentiated Niacin activation (left) and dose-response curves of Niacin and compound 9n alone induced Gi1 signaling (right).

Reviewer #3

Mao et al. presenta manuscript describing the binding mode of endogenous and synthetic ligands to HCAR2, a receptor involved in the metabolism of hydrocarboxylic acids and a target for the treatment of dyslipidemia. The authors present three cryo-EM structures of HCAR2 coupled to a Gi heterotrimer and bound to the endogenous 3-HB, Niacin and a synthetic MK-6892, where an allosteric compound (compound 9n) is additionally present at the 3-HB and Niacin structures. The authors show functional and computational data supporting the binding modes seen in the structural data. The manuscript is well-written and organized, and present relevant data. Although, the recent publication of HCAR2 structures has an impact on the novelty of the work, the authors present a significant amount of additional data and functional assays. In my opinion the manuscript is recommended for publication with minor modifications:

Response: We are grateful for the referee's positive comments on our study.

1. The challenge of studying such small agonists is being certain about the binding pose of such ligands within the cryo-EM maps, which are generally of lower resolution when compared to x-ray structures. In this case, the authors manage to achieve relatively high resolution and support the binding mode of 3-HB and Niacin with docking and MD simulations. However, the authors should acknowledge that, based on experimental maps, Niacin and 3-HB could be fitted the opposite orientation within the map. Alternatively, the authors should elaborate slightly more to defend the chosen orientation of both ligands.

Response: We express our appreciation for the referee's insightful comments. Indeed, due to the small size of both 3-HB and Niacin, along with their nearly identical opposite orientations, distinguishing between these two binding modes solely based on cryo-EM density presents a considerable challenge. Detailed structural analysis has demonstrated that the interacting residues in these two binding modes are almost identical. The primary divergence lies in the hydroxyl group of 3-HB and the pyridinic nitrogen of Niacin, as well as their interactions with the receptor. In our current structure, the hydroxyl group of 3-HB can form hydrogen bonds with both the main chain and side chain of S179^{ECL2} within the receptor. Conversely, in its opposite orientation, only a hydrogen bond can be established with Y284^{7.43}. As for Niacin, its two distinct poses can each form hydrogen bonds with the main chain of S179^{ECL2} and the side chain of

Y87^{2.64}. Subsequent functional analysis revealed that the Y87^{2.64}F mutation significantly reduced Niacin's activation (Supplementary Fig.7c). Therefore, based on our structure and functional findings, we adopted the binding pose of Niacin that forms a hydrogen bond with Y87^{2.64}. Consistent with our structure and functional analysis, our docking analysis have indicated that the top two binding poses are in agreement with our current structure for both 3-HB and Niacin, with the third pose adopting the opposite orientation (Supplementary Fig.7a). Therefore, we reason that our structure represents the favorable binding pose for 3-HB and Niacin. We have included corresponding descriptions and Figures within our revised manuscript (Supplementary Fig.7e and Lines 119-122).

Supplementary Fig.7 | Recognition of 3-HB and Niacin by HCAR2.

e Two potential binding poses of Niacin and 3-HB. Structural analysis demonstrates that the interacting residues in the two binding modes of the two ligands are almost identical. The primary divergence lies in the hydroxyl group of 3-HB and the pyridinic nitrogen of Niacin, as well as their interactions with the receptor. In our current structure, the hydroxyl group of 3-HB can form hydrogen bonds with both the main chain and side chain of S179^{ECL2} within the receptor. Conversely, in its opposite orientation, only a hydrogen bond can be established with Y284^{7.43}. For Niacin, its two distinct poses can each form hydrogen bonds with the main chain of S179^{ECL2} and the side chain of Y87^{2.64}. Further functional analysis reveals that the Y87^{2.64}F mutation significantly reduced Niacin's activation.

2. The authors comment separately on the binding modes on 3-HB and Niacin and compound9, however the structures of 3-HB and Niacin in the absence of compound9 have not been obtained, hence the authors should include a comment about the potential influence on the allosteric compound9 on the binding mode of 3-HB and Niacin.

Response: We thank the referee for his/her valuable comments. Our structural analysis indicated that compound 9n does not directly interact with either the ligand or the residues involved in ligand binding. It is likely that compound 9n exerts its PAM effect by stabilizing the active conformation of the receptor, thereby enhancing the stability of agonist binding, as elucidated by our MD simulations. Consequently, it is probable that Compound 9n exerts either no or only minimal influence on the recognition mode of the orthosteric agonists (Supplementary Fig.10f). We agree with the reviewer's perspective and to address the potential concerns of misinterpretation, the following statement has been included in the discussion section (Lines 312-317): "Our analysis of the recognition modes of the orthosteric agonist and compound 9n is based on the

structures involving simultaneous binding of both ligands. Despite their distinct binding pockets and absence of direct interactions, future investigations involving HCAR2 complexed with 3-HB, Niacin, and compound 9n alone will offer a more comprehensive understanding of the ligand recognition of HCAR2."

Supplementary Fig.10 | Recognition of compound 9n by HCAR2.

f Frequency distribution plot of agonists centroids with or without compound 9n (light green). Frequency distribution is colored by spectrum from high (red) to low (blue), initial positions of agonists are shown in magenta.

- 1 Yang, Y. *et al.* Structural insights into the human niacin receptor HCA2-G(i) signalling complex. *Nat Commun* **14**, 1692, doi:10.1038/s41467-023-37177-6 (2023).
- 2 Darden, T., York, D. & Pedersen, L. PARTICLE MESH EWALD - AN N.LOG(N) METHOD FOR EWALD SUMS IN LARGE SYSTEMS. *Journal of Chemical Physics* **98**, 10089-10092, doi:10.1063/1.464397 (1993).
- 3 Lauring, B. *et al.* Niacin lipid efficacy is independent of both the niacin receptor GPR109A and free fatty acid suppression. *Sci Transl Med* **4**, 148ra115, doi:10.1126/scitranslmed.3003877 (2012).
- 4 Offermanns, S. It ain't over 'til the fat lady sings. *Sci Transl Med* **4**, 148fs130, doi:10.1126/scitranslmed.3004445 (2012).
- 5 Shen, H. C. *et al.* Discovery of pyrazolopyrimidines as the first class of allosteric agonists for the high affinity nicotinic acid receptor GPR109A. *Bioorg Med Chem Lett* **18**, 4948-4951, doi:10.1016/j.bmcl.2008.08.039 (2008).

- 6 Xu, J. *et al.* Structural and dynamic insights into supra-physiological activation
and allosteric modulation of a muscarinic acetylcholine receptor. *Nat Commun*
14, 376, doi:10.1038/s41467-022-35726-z (2023).
- 7 Schwartz, T. W. & Holst, B. Allosteric enhancers, allosteric agonists and ago-
allosteric modulators: where do they bind and how do they act? *Trends*
Pharmacol Sci **28**, 366-373, doi:10.1016/j.tips.2007.06.008 (2007).
- 8 Draper-Joyce, C. J. *et al.* Positive allosteric mechanisms of adenosine A(1)
receptor-mediated analgesia. *Nature* **597**, 571-576, doi:10.1038/s41586-021-
03897-2 (2021).
- 9 Chan, W. Y. *et al.* Allosteric modulation of the muscarinic M4 receptor as an
approach to treating schizophrenia. *Proc Natl Acad Sci U S A* **105**, 10978-10983,
doi:10.1073/pnas.0800567105 (2008).
- 10 Kew, J. N. Positive and negative allosteric modulation of metabotropic
glutamate receptors: emerging therapeutic potential. *Pharmacol Ther* **104**, 233-
244, doi:10.1016/j.pharmthera.2004.08.010 (2004).
- 11 Bhattacharya, S. & Vaidehi, N. Differences in allosteric communication
pipelines in the inactive and active states of a GPCR. *Biophys J* **107**, 422-434,
doi:10.1016/j.bpj.2014.06.015 (2014).
- 12 DeVree, B. T. *et al.* Allosteric coupling from G protein to the agonist-binding
pocket in GPCRs. *Nature* **535**, 182-186, doi:10.1038/nature18324 (2016).
- 13 Holst, B., Elling, C. E. & Schwartz, T. W. Partial agonism through a zinc-Ion
switch constructed between transmembrane domains III and VII in the
tachykinin NK(1) receptor. *Mol Pharmacol* **58**, 263-270,
doi:10.1124/mol.58.2.263 (2000).

REVIEWERS' COMMENTS

Reviewer #1 (Remarks to the Author):

All my comments were properly answered and the additional required simulations and analyses were performed. I have no more comments.

Reviewer #2 (Remarks to the Author):

The Paper is OK now

Reviewer #3 (Remarks to the Author):

The authors have made a good effort to satisfy all concerns addressed. I have no further concerns.

Responses to the reviewers' comments

We thank the referees for their valuable time reviewing our manuscript and the constructive suggestions that they have provided. We have carefully taken these comments into consideration when preparing a revision, which resulted in a more thorough and clear manuscript. We have copied each comment in *Black Italic*, which is followed by our own point-by-point response in **Blue**, including details about the corresponding changes to the manuscript.

Reviewer #1

All my comments were properly answered and the additional required simulations and analyses were performed. I have no more comments.

Response: We thank the reviewer for the positive comments and concerns on our manuscript.

Reviewer #2

The Paper is OK now.

Response: We are grateful to the reviewer's concerns of our study.

Reviewer #3

The authors have made a good effort to satisfy all concerns addressed. I have no further concerns.

Response: We thank again for the referee's questions and concerns.